# Triple-Isotope Tracing for Pathway Discernment of NMN-Induced NAD^+^ Biosynthesis in Whole Mice

**DOI:** 10.3390/ijms241311114

**Published:** 2023-07-05

**Authors:** Anthony A. Sauve, Qinghui Wang, Ning Zhang, Seolhee Kang, Abigail Rathmann, Yue Yang

**Affiliations:** Department of Pharmacology, Weill Cornell Medicine, 1300 York Avenue, New York, NY 10065, USA

**Keywords:** NAD^+^ metabolism, NMN, NR, nicotinamide, pharmacodynamics

## Abstract

Numerous efforts in basic and clinical studies have explored the potential anti-aging and health-promoting effects of NAD^+^-boosting compounds such as nicotinamide riboside (NR) and nicotinamide mononucleotide (NMN). Despite these extensive efforts, our understanding and characterization of their whole-body pharmacodynamics, impact on NAD^+^ tissue distribution, and mechanism of action in various tissues remain incomplete. In this study, we administered NMN via intraperitoneal injection or oral gavage and conducted a rigorous evaluation of NMN’s pharmacodynamic effects on whole-body NAD^+^ homeostasis in mice. To provide more confident insights into NMN metabolism and NAD^+^ biosynthesis across different tissues and organs, we employed a novel approach using triple-isotopically labeled [^18^O-phosphoryl-^18^O-carbonyl-^13^C-1-ribosyl] NMN. Our results provide a more comprehensive characterization of the NMN impact on NAD^+^ concentrations and absolute amounts in various tissues and the whole body. We also demonstrate that mice primarily rely on the nicotinamide and NR salvage pathways to generate NAD^+^ from NMN, while the uptake of intact NMN plays a minimal role. Overall, the tissue-specific pharmacodynamic effects of NMN administration through different routes offer novel insights into whole-body NAD^+^ homeostasis, laying a crucial foundation for the development of NMN as a therapeutic supplement in humans.

## 1. Introductions

NAD^+^ is a redox-active metabolite that plays a crucial role in the catabolism of fuel substrates in the body [1]. Additionally, NAD^+^ is an integral component of cellular adaptation mechanisms, as it serves as a substrate for sirtuins and poly-ADP-ribosyl polymerases [1,2,3]. Enhancing NAD^+^ levels has been linked to various biological effects, including mitochondrial biogenesis [4,5], improved insulin sensitivity [4,5], resistance to obesity [4,5], protection against cell death [6,7,8], optimization of mitochondrial performance [9], rejuvenation [9,10,11], resistance to neurodegenerative disease [12,13,14], and increased lifespan and/or health span in mice [15,16,17]. These biological consequences have led to numerous clinical studies of pharmacological agents that increase whole-body NAD^+^ [6,7,18]. However, despite extensive preclinical and clinical investigation of these NAD^+^ boosters, the mechanisms by which they increase tissue NAD^+^ in mice and humans remain poorly understood and, in some cases, controversial [19,20].

Nicotinamide riboside (NR) and nicotinamide mononucleotide (NMN) are two NAD^+^ boosters that have been extensively studied for their ability to modulate NAD^+^ concentrations in mammalian models of disease and aging. The majority of NAD^+^ synthesis following oral intake of NR or NMN has been attributed to the breakdown of these precursors to nicotinamide, which is then salvaged via nicotinamide phosphoribosyltransferase (Nampt) to form NMN and later converted to NAD^+^ via the actions of adenylyltransferases (Nmnat1,2,3) [6,7,21]. Recent works have also shown that a significant proportion of NR is cleaved by purine nucleoside phosphorylase (PNP) intracellularly, leading to the accumulation of nicotinamide [22]. Additionally, NR is mainly processed via nicotinamide riboside kinase 1 (Nrk1) [6,7,21], which phosphorylates the 5′-OH group of NR and produces NMN as the intracellular substrate for NAD^+^ synthesis.

In contrast, the pathways for pharmacologically administered NMN-induced NAD^+^ synthesis are less clear. Intact NMN can potentially be transported across cell membranes, and a putative NMN transporter called Slc12a8 has been reported to be important for the in vivo action of NMN [23]. Slc12a8 is highly expressed in the jejunum, ileum, and pancreas and moderately expressed in the liver and white adipose tissues. Overexpression of Slc12a8 facilitates the uptake of radio- and stable-isotope-labeled NMN in cultured NIH 3T3 cells [21]. However, the absence of labeling on the phosphate group of NMN did not allow researchers to answer the important question of whether NMN was dephosphorylated prior to cell entry, challenging the existence of this NMN transporter [19,23]. On the other hand, other studies suggest that NMN requires extracellular dephosphorylation into NR before entering cells, as Nrk1 overexpression was required for NMN-induced increases in NAD^+^ concentrations in NIH 3T3 cells [21]. Nrk1 knockout primary hepatocytes from mice exhibited diminished synthesis of NAD^+^ from NMN and NR, and Nrk1KO mice had a blunted response to intraperitoneal (IP) injection of NMN compared to wild-type mice, indicating a requirement for the Nrk1 enzyme to complete NAD^+^ biosynthesis [21]. However, another study employing isotope labeling of NR moieties reported less than 10% incorporation of the labeled NR moiety from intravenous (IV) injected NMN and no formation of NR-labeled NAD^+^ from oral gavage delivery [24], indicating that NMN-induced NAD^+^ synthesis was independent of Nrk1. Due to the ambiguity in labeling, the question of how NMN is metabolized in vivo is still unsettled.

Our study aimed to expand the current understanding of tissue NAD^+^ homeostasis in mice by investigating the impact of NMN on NAD^+^ concentrations and absolute amounts in various tissues and the whole body. To clarify the metabolic fate of NMN in vivo, we synthesized a novel isotopic tool, [^18^O-phosphoryl-^18^O-carbonyl-^13^C-1-ribosyl] NMN, which enabled us to directly assess the incorporation of the entire NMN molecule with the labeled phosphate moiety. We were able to quantify the relative contributions of nicotinamide salvage, NR salvage, and intact NMN incorporation pathways to NAD^+^ biosynthesis. Our study provides a comprehensive characterization of NAD^+^ homeostasis in mice and a precise quantification of the pharmacological effects of NMN.

## 2. Results

### 2.1. NAD^+^ Concentrations in Mouse Tissues

To understand whole-body NAD^+^ homeostasis in vivo, we first measured the absolute NAD^+^ concentrations and total NAD^+^ in various tissues in 10-week-old male C57BL/6N mice fed a standard chow diet (Figure 1A and Table 1). Young, male mice were selected for this study to minimize the potential impacts of hormonal variations on NAD^+^ metabolism. Among the tissues tested, the kidney (736 ± 61 pmol/mg tissue) and liver (621 ± 81 pmol/mg) exhibited the highest NAD^+^ concentrations, consistent with previous reports [6,25,26]. The heart (297 ± 83 pmol/mg) had the highest NAD^+^ concentration of the remaining tissues. This may be attributed to the high rate of energy consumption and the greater abundance of mitochondrial-specific enzyme activity found in the heart [27]. For instance, the heart contains two to six times more cytochrome oxidase and citrate synthase activities than the liver or kidney [27]. However, the ranking order of NAD^+^ concentrations in mouse tissues cannot be solely predicted by increased mitochondrial activity. The NAD^+^ concentrations in the lung, brain, brown adipose tissue, skeletal muscle, pancreas, stomach, spleen (Figure 1A), and periphery (carcass minus organs, Table 1) ranged from 100 to 200 pmol/mg tissue. The lowest measured NAD^+^ concentrations were found in whole blood (32 ± 5 pmol/mg) and white adipose tissue (15 ± 6 pmol/mg). Evidently, NAD^+^ concentrations are tissue-specific and can vary up to 30-fold.

### 2.2. Total Tissue NAD^+^ Amounts in the Whole Body of Young Male Mice

To fully understand whole-body NAD^+^ homeostasis, it is important to account for not only tissue concentrations but also total NAD^+^ content in each tissue, which is an aspect largely overlooked in previous literature. In our study, we recorded the weights for whole tissues except for skeletal muscle and white adipose tissues, where we sampled only quadriceps femoris muscles and epididymal adipose pads. The liver, with its larger mass and relatively high NAD^+^ concentration, contains the highest NAD^+^ content among individual organs, averaging 812 ± 109 nmol/per liver (Figure 1B and Table 1). The kidneys are smaller in size, and a pair contained an average of 242 ± 40 nmol NAD^+^. The combined NAD^+^ content of the kidney and liver accounts for approximately a quarter of the total NAD^+^ content (Table 1). The remaining internal organs, excluding the liver, kidney, and carcass, have a total NAD^+^ of approximately 400 nmol (Table 1). The highest NAD^+^ contents among these organs are found in the intestine (84 ± 25 nmol), skeletal muscle/quadriceps (61 ± 31 nmol), brain (58 ± 3 nmol), blood (47 ± 10 nmol), stomach (39 ± 5 nmol), and lung (28 ± 2 nmol, Figure 1B and Table 1). The NAD^+^ content of the remaining carcass, which mainly consists of skin, fur, bone, and connective tissues, was estimated to be 3178 ± 325 nmol (Table 1). In sum, the total combined NAD^+^ content in the whole body was 4638 ± 582 nmol in control mice.

### 2.3. NAD^+^ Concentrations and Contents with NMN IP Administration

We then tested the impact of IP-injected NMN on tissue NAD^+^ concentrations and contents. At 4 h post-IP injection of 500 mg/kg NMN, the concentrations of NAD^+^ in the liver and kidney were increased by approximately 2.4-fold to levels of 1493 ±121 pmol/mg and 1761 ± 23 pmol/mg, respectively (Figure 1C,E). The epididymal white adipose tissue showed a two-fold increase in NAD^+^ concentrations. NAD^+^ concentrations in the pancreas also showed a substantial response (1.7-fold), followed by the heart (1.5-fold) and the lung (1.2-fold). (Figure 1C). Peripheral NAD^+^ measured from the remaining carcass was similar to controls (Table 1).

Total NAD^+^ abundances in tissues were also enriched by NMN IP injection. Assuming the injected NMN is completely absorbed and converted to NAD^+^, the maximum potential increase in total NAD^+^ content in the body of a 25-g mouse would be 24.8 mg (37,313 nmol). This is approximately eight times higher than the measured average whole-body NAD^+^ content (4638 nmol) in control animals (Table 1). However, the whole-body NAD^+^ content in NMN IP-injected mice was calculated to be 6225 ± 726 nmol, only 1587 nmol more than controls. Liver and kidney NAD^+^ content both increased by 2.5-fold, accounting for 98% (1559 nmol) of the total NAD^+^ increases (Figure 1D,E, Table 1). This indicates that, compared to other tissues, the liver and kidney are more responsive to NMN IP injections. In contrast, all the other organs combined only contributed to a 2% increase in total NAD^+^ content. The peripheral carcass, although containing the majority of NAD^+^, was not responsive to NMN injection at this time point (Table 1). Based on our data, the pharmacodynamic effect of NMN observed is about 4% of its maximal possible impact, suggesting a low absorption and conversion rate of NMN overall.

### 2.4. NAD^+^ Concentrations and Contents with NMN Gavage Administration

The effects of oral gavage administration of NMN on NAD^+^ concentrations were also investigated. Compared to IP injection, the NAD^+^ enhancement at 4 h post-gavage of 500 mg/kg NMN was generally less effective. The liver was an exception and showed the most significant increase in concentrations, reaching 1413 ± 333 pmol/mg, a 2.3-fold increase versus control (Figure 1F), which is comparable to the effect of IP injection. In contrast, the kidney NAD^+^ concentrations were only increased by 1.3-fold over controls, substantially lower than with IP injection. The epidydimal white adipose tissue, pancreas, and lung showed increases in NAD^+^ concentrations of 1.8-, 1.7-, and 1.2-fold over controls, respectively (Figure 1F, Table 1). Although orally administered NMN has been reported to boost NAD^+^ levels in muscles [9], we observed negligible effects on NAD^+^ in skeletal muscle with single-dose gavaged NMN (Figure 1F,H, Table 1).

Overall, oral gavage provided a lower impact on whole-body NAD^+^ contents at 4 h compared to IP injection. There is also the potential for more excretion through feces following the gavage of NMN. Total NAD^+^ content from all tissues amounted to 5163 ± 693 nmol, which was only 525 nmol more than controls. The liver alone was responsible for 875 nmol of the increase, while all other measured internal organs combined contributed to a total of 185 nmol increases in NAD^+^ content (Figure 1G, Table 1). Nevertheless, the total NAD^+^ content of the peripheral carcass was reduced to 2644 nmol, a reduction of 17% versus untreated controls (Table 1).

### 2.5. Time Dependence for NAD^+^ Increases

IP injection enables more rapid absorption of NMN compared to oral gavage; therefore, the differences in tissue NAD^+^ enhancements may be related to the timing of collection. To evaluate the effects of time, we dosed mice with 500 mg/kg of NMN by IP or gavage and sacrificed them at 2, 4, and 6 h. Based on previous investigations indicating that the pharmacodynamic effect of NR peaked at 6 h post gavage in mice [28], longer time points beyond 6 h were not selected for this study. Substantial NAD^+^ enhancements were observed at 2 h in the liver, kidney, blood, small intestine, and epidydimal white adipose tissues following IP injection. However, only the small intestine exhibited significant NAD^+^ increases at 2 hours post-treatment in oral-gavaged mice, which is not surprising as NMN reaches the gastrointestinal (GI) tracts first, indicating a lag in the uptake of NMN and its biometabolites into tissues with gavage administration (Figure 2). In both IP and gavage groups, the highest NAD^+^ concentrations in the liver, kidney, and pancreas were observed between 4 and 6 h. However, this effect was not duplicated in all tissues. In the IP-injected group, white adipose tissue had a rapid response and peaked at 2 h, while the heart showed maximal NAD^+^ contents at 4 h. In the blood, small intestine, brown adipose, and lung, higher concentrations were reached at 6 h. In the gavaged group, the blood and small intestine exhibited the highest NAD^+^ concentrations at 6 h. We observed no significant difference in the NAD^+^ amount in peripheral tissues at 6 h, although when sampling the peritoneal membrane as a specific part of peripheral tissues, we found an elevation in NAD^+^ levels that peaked at 4 h following IP administration. The different pharmacodynamic impacts of administered NMN likely reflect preferential redistributions of NMN and its biometabolites after the original bolus was absorbed and metabolized by the liver (IP) or the digestive system (gavage). We, therefore, suggest that the lower degrees of NAD^+^ enhancement observed in most tissues with oral gavage compared to IP were not related to the time of tissue collection but rather to the route of administration and excretion.

### 2.6. Isotope Labeling to Assess Pathways of Synthesis

To investigate the NAD^+^ biosynthetic pathways involved in various tissues, we then utilized a synthetic, triple-labeled NMN. This labeled NMN is distinct from the ones used in previous investigations because it features separate labeling on the phosphate, sugar, and base moieties, allowing for a comprehensive tracing of their metabolic fate. The incorporation of ^18^O in the phosphate moiety (95% incorporation), ^13^C in C1’ of the ribose ring (99% incorporation), and ^18^O in the carbonyl nicotinamide (90% incorporation) enables confirmative tracing of the metabolic fate of each moiety. The three major pathways for NAD^+^ biosynthesis that can be used by NMN are illustrated in Figure 3, and the impact of each metabolic pathway on the relative *m*/*z* ratios of the NAD^+^ produced is indicated. The first two pathways may arise from the metabolic decomposition of NMN before the biosynthesis of NAD^+^. Pathway 1 involves the breakdown of NMN to nicotinamide, catalyzed by the ecto-enzyme CD38 [29], before entering cells. Also included in this pathway is the nicotinamide generated from the intracellular cleavage of NR by PNP. Tissues utilizing this labeled nicotinamide as a precursor generate NAD^+^ with two additional mass units (*m*/*z* = 666) compared to the control (*m*/*z* = 664). In Pathway 2, NMN is decomposed by an unknown phosphatase to form NR, which is then incorporated into NAD^+^ via the action of Nrk1/2 to reform NMN in cells. NAD^+^ produced via Pathway 2 retains both ^13^C-ribose and ^18^O-nicotinamide, resulting in a product with *m*/*z* = 667. Finally, the transport of NMN across membranes results in the incorporation of fully labeled NMN into NAD^+^, and the corresponding NAD^+^ is proposed to have five additional mass units (*m*/*z* = 669) via Pathway 3. The isotopic label in the phosphate allows the detection of this pathway in tissue samples.

### 2.7. IP Administration of Isotope-Labeled NMN

We first administered [^18^O-phosphoryl-^18^O-carbonyl-^13^C-1-ribosyl] NMN by IP injection of 500 mg/kg into 10-week-old C57BL/6N male mice and sacrificed them after 2 or 4 h (*n* = 3). We subtracted average NAD^+^ mass intensities at 666, 667, and 669 from control mice to reflect true incorporations of isotopic material. The adjusted intensities are expressed against 664 as 100% intensity (Figure 4). In the liver, where we observed the most NAD^+^ enhancement, about 25% and 17% of isotopic intensities at *m*/*z* = 666 reflect nicotinamide incorporation from Pathway 1, and a comparable 30% and 13% intensity at *m*/*z* = 667 reflect NR incorporation from Pathway 2, at the 2- and 4-hour time points, respectively. Kidney exhibited a different pattern, with 20% and 19% labeling from Pathway 1 (*m*/*z* = 666), and 86% and 47% signal intensities from Pathway 2 (*m*/*z* = 667) at 2 and 4 h, respectively. This finding indicates that kidney NAD^+^ production is dominated by isotopically labeled NR incorporation into NAD^+^. Blood exhibited very weak *m*/*z* = 667 labeling, suggesting that all NAD^+^ enhancements in the blood observed are most likely traceable to nicotinamide salvage. Almost all other tissues examined utilized both the nicotinamide and NR salvage pathways. Spleen exhibited strikingly high *m*/*z* = 666 (75–104%) and high *m*/*z* = 667 (20–38%) intensities, indicating a high utilization rate with labeled nicotinamide and NR for NAD^+^ synthesis (Figure 4).

The intensities of the *m*/*z* = 669 peak were at or below the limit of detection in all cases except for kidney and white adipose tissues, where adjusted amounts were low but evident at approximately 4–6% and 3–10%, respectively (Figure 4). It is worth noting that the putative NMN transporter was found to be expressed at moderate levels in white adipose tissues [20]. Additionally, although there are no reports of the expression of Slc12a8 in the kidney, other members of the Slc12 family have been identified in the kidney [30], supporting the existence of a potential NMN transporter in these tissues. We, therefore, conclude that the transport of intact NMN likely occurs in small quantities in the kidney and white adipose tissues following IP injection.

### 2.8. Gavage Administration of Isotope

We also administered triple-isotope-labeled NMN by gavage at 500 mg/kg for 2 and 4 h (Figure 5). In all tissues, the dominant pathway for NAD^+^ production was through nicotinamide salvage, as indicated by the *m*/*z* = 666 signals. In the liver, we observed a delayed appearance of *m*/*z* = 666, with 5% intensity at 2 h increasing to 15% at 4 h, which was also reflected in the delayed time to reach maximum NAD^+^ enhancement. Similarly, delayed maximum nicotinamide incorporations were observed at 4 h in all tissues, ranging from five to seventeen percent, except for blood. In the spleen, nicotinamide incorporation reached nearly 100% of the endogenous 664 signal, and NR incorporation reached 24% at 4 h. In addition, the liver, pancreas, kidney, white and brown adipose tissues, and brain all exhibited very small enhancements in NR-induced *m*/*z* = 667 peaks, not exceeding 10%. In addition, we did not find clear evidence for appreciable NMN incorporation following oral administration except for a 7% *m*/*z* = 669 signal in the epidydimal fat pad at 2 h, further confirming the existence of NMN transporters in white adipose tissues (Figure 5).

### 2.9. Intestinal NAD^+^ Biosynthesis with Isotopic Labels

To further investigate the transport of intact NMN across cell membranes, we measured the isotope incorporation in NAD^+^ in the small intestine, where Slc12a8 is highly expressed. For the isotopic analysis, we divided the small intestine into four segments, roughly representing the duodenum (Proximal intestine-1), jejunum (Proximal Intestine-2), and ileum (Distal Intestine-2 and Distal Intestine-1, which are adjacent to the cecum). Our findings showed that both administration routes lead to 1.3–1.6 fold increases in intestinal NAD^+^ concentrations between 2 and 4 h (Figure 2). Isotopic NAD^+^ with *m*/*z* = 666 increased by 31–128% at 2 and 4 h in all segments of the intestine after IP injection and by 24–62% after gavage, indicating the active absorption of labeled nicotinamide for NAD^+^ synthesis (Figure 6). Moreover, uptake of labeled NR, represented by *m*/*z* = 667 intensities, was increased by over 50% at 2 h in all segments of IP-injected intestines. In gavaged mice, the *m*/*z* = 667 intensities were elevated by 17–31% in the upper intestinal segments but were only elevated by 12–14% in the lower intestinal segments at the 2-hour time point (Figure 6). Across all segments, there was a decrease in isotope labeling at 4 h versus 2 h, with the only exception being the jejunum, where the labeled NAD^+^ intensities were higher at 4 h after gavage. Our findings suggest that both nicotinamide and NR can enter enterocytes from both the basolateral side (IP) and the luminal side (gavage) and then undergo conversion to NAD^+^, implying that the intestines play a crucial role in the metabolic absorption and metabolism of NAD^+^ precursors to NAD^+^. Nevertheless, we observed a neglectable amount of NMN uptake, indicated by less than 1% *m*/*z* = 669 intensities, in the intestine after either administration route (Figure 6), which is not in agreement with the high expression of the putative NMN transporter Slc12a8 in the jejunum and ileum [20]. Overall, in the small intestines, we observed the highest incorporation of isotope signals from both nicotinamide and NR with only moderate increases in total NAD^+^ content, implying rapid NAD^+^ turnover in the small intestine elicited by NMN administration.

### 2.10. Nicotinamide in Whole Blood with NMN IP and Gavage Administration

Based on our findings, we deduce that the majority of new NAD^+^ synthesis comes from the incorporation of labeled nicotinamide in most tissues. To further investigate this, we measured nicotinamide levels in the blood of isotopically labeled NMN-administered mice. As shown in Figure 7A, an IP injection of 500 mg/kg NMN resulted in a significant increase in blood nicotinamide levels, reaching over 16-fold at 2 h post-treatment, from 137.2 ± 35.9 pmol/mg blood (Ctrl) to 2302.9 ± 160.9 pmol/mg. This effect was transient, decreasing to 48.2 ± 45.8 pmol/mg at 4 h. At 2 h, ^18^O-nicotinamide (*m*/*z* = 124, +2) accounted for 81% of total blood nicotinamide and was 4.3-fold higher than unlabeled nicotinamide (*m*/*z* = 122). At 4 h, the level of labeled nicotinamide was decreased to 48% of total nicotinamide and was only 1.1-fold over unlabeled (Figure 7B), which is in accordance with the lower *m*/*z* = 666 labeling at 4 h versus 2 h seen in the majority of tissues following IP injection in Figure 4.

In contrast, gavaged NMN resulted in a gradual increase in blood nicotinamide concentrations, reaching a 1.9-fold increase to 260.3 ± 28.1 pmol/mg blood at 2 h and a 2.9-fold increase to 402.1 ± 53.6 pmol/mg at 4 h (Figure 7A). At 2 and 4 h, the intensities of ^18^O-labeled nicotinamide were at 37% and 40%, respectively, of unlabeled nicotinamide, indicating that most of the increased nicotinamide in the circulation was unlabeled (Figure 7B). These results not only suggest a lower bioavailability of NMN after oral gavage but also imply an enhanced turnover of NAD^+^ and the release of nicotinamide from tissues.

## 3. Discussion

A recent FDA ruling has reclassified NMN as a potential drug instead of a dietary supplement [31]. Indeed, although NMN has been extensively studied for its therapeutic benefits in a wide range of diseases in both humans and mice, its metabolic fate in vivo remains unclear. In this study, we present a more complete characterization of the pharmacodynamic properties of NMN in mice. For the first time, we report changes in NAD^+^ pools within various previously untested tissues, such as adipose tissues, the small intestine, and peripheral tissues. To track the incorporation of intact NMN molecules into NAD^+^ synthesis in vivo, we utilized state-of-the-art techniques to produce a novel NMN compound labeled with isotopes on the phosphate, ribose, and nicotinamide moieties. This discovery could provide new insights into the in vivo metabolism of NAD^+^ throughout the body.

Two commonly used administrative routes were tested in this study. The administration of NMN through IP injection allowed for rapid delivery and a strong enhancement of NAD^+^ levels, while oral gavage results in a slower delivery with less impact on NAD^+^ metabolism. Data from isotope-labeled compounds revealed that the nicotinamide salvage pathway for NAD^+^ biosynthesis was the dominant pathway in almost all tissues except for the kidney. This was also supported by the surge of both labeled and total nicotinamide concentrations in the blood 2 h post-IP injection and 2–4 h post-gavage (Figure 7A,B). The incorporation of NR for NAD^+^ biosynthesis was also prominent in most tissues following IP injection, but most evident in the kidney and small intestines. In the kidney, more than 40% of total NAD^+^ was synthesized from isotope-labeled NR 4 h post-IP injection (Figure 7C), which may be attributed to the kidney’s role in catabolizing excess NMN and its dephosphorylated products, NR. We also found a small amount of direct incorporation of intact NMN in the kidney and epidydimal white adipose, suggesting that NMN can be transported into these organs via a transporter and converted to NAD^+^ (Figure 7C). However, we did not observe similar intact NMN incorporation in other tissues reported to express Slc12a8, e.g., the small intestine, liver, and pancreas. Therefore, it is still unclear whether this transporter is Slc12a8, and further research is required to determine if Slc12a8 is indispensable for the incorporation of intact NMN in kidney and white adipose tissues.

By comparing the changes in total NAD^+^ and isotopically labeled NAD^+^, we found that metabolically active tissues such as the liver utilize a considerable amount of unlabeled nicotinamide as an NAD^+^ precursor following NMN administration. The liver displayed the most significant increase in NAD^+^ levels after both routes of NMN administration (Figure 2 and Figure 3). Four hours after the IP injection, the liver exhibited a 2.5-fold increase in total NAD^+^ levels. Theoretically, if all the increased NAD^+^ came from an isotope-labeled NMN, the *m*/*z* = 666 (+2) and 667 (+3) signal intensities should have been close to 60% of all NAD^+^. However, the results showed less than 23% of total NAD^+^ was labeled with isotopes at 4 h (see pie chart in Figure 7C). These findings suggest that even though the liver is the first organ to metabolize IP-injected NMN, it is actively breaking down the newly administered compounds and releasing them into the blood while preferentially salvaging the circulating nicotinamide for its own NAD^+^ biosynthesis. In the livers of orally gavaged mice, the total NAD^+^ increase was similar, but the combined labeling percentage was even lower, at approximately 17% (Figure 7D). These findings align with those of Liu et al., who demonstrated weak signals for isotope incorporation in the liver from both NR and NMN delivered by gavage or IV at 500 mg/kg [24].

On the contrary, while the total amount of NAD^+^ in the small intestines showed only moderate increases, these NAD^+^ molecules were substantially labeled with isotopes. In mice IP injected with NMN, after 4 h, we detected an average of 47% of total NAD^+^ labeled with isotopes (Figure 7C). If all of the newly synthesized NAD^+^ represented a net increase, the small intestines should have an increase of 1.9-fold over the control in their total NAD^+^ amount. However, the NAD^+^ levels increased only 1.5-fold, indicating a 20% discrepancy in the NAD^+^ pool. Similarly, oral gavage of NMN also revealed an active turnover of NAD^+^ in the intestines, with a high labeling rate of 35% of NAD^+^ (equivalent to a 1.5-fold increase) and a small 1.3-fold increase in NAD^+^ content (Figure 7D). We, therefore, deduce that a considerable portion of the unlabeled NAD^+^ content in the gastrointestinal tract was turned over rapidly upon NMN administration, releasing unlabeled nicotinamide into the blood.

In further support of our speculation about the redistribution of NAD^+^ and nicotinamide, we also found an increase in blood nicotinamide concentrations 4 h after NMN oral gavage, but only around 40% of the nicotinamide was labeled. Another potential source for releasing the unlabeled nicotinamide may be the peripheral tissues, as we noted a 20% decrease in peripheral NAD^+^ contents at 4 h, roughly equal to 535 nmol NAD^+^ (Table 1). At 4 h, 20% of liver NAD^+^ (330 nmol) was labeled, which, when combined with the 535 nmol NAD^+^ deducted from the periphery, accounted for the total increase of 800 nmol. We therefore speculate that within 4 h after NMN oral administration, there is a release of nicotinamide from both the intestines and peripheral tissues, which, in combination with isotopically labeled nicotinamide, is used by the liver for new NAD^+^ production (Figure 7D). The mechanism of this process is not yet understood, but it is possible that a physiologic signaling mechanism, such as a hormone, a neuronal-mediated signal, or both, accounts for enhanced NAD^+^ turnover in the periphery.

We also considered the possibility that some NAD^+^ originates from NMN metabolites produced by the microbiome in the gastrointestinal tract, particularly in mice that were orally administered NMN; however, we believe that bacterial contribution had a minimal impact on the interpretation of the results. Previous studies on NR administration have reported the accumulation of nicotinic acid (NA)-derived metabolites in tissues [32]. Since the host genome does not encode enzymes that produce NA-related metabolites, it is likely that these metabolites originate from microbiome-derived enzymes such as nicotinamidase. NAD^+^ synthesis through NA would result in at least a 50% loss of the ^18^O-carbonyl label in our isotopically labeled material, assuming NA is not subject to additional turnover in the gut. However, data from Liu et al. suggest that NA is unlikely to be a major contributor to NAD^+^ synthesis through the oral route. They found that the labeling of NAD^+^ in the liver from gavage administration of 50 mg/kg NMN did not exceed 15% of total NAD^+^. Incorporation of isotopes after oral administration of NR at 200 mg/kg also remained low, not exceeding 20% of total NAD^+^, and single ^13^C label incorporation that remains with NA did not exceed 15% [24]. These numbers are similar to what we have observed (12% isotope labeling at 2 h and 17% at 4 h) after NMN gavage. Therefore, we do not believe that the low percentage of labeling is explained by a prior conversion to NA, even though NMN is likely converted to NA-related metabolites in the gut. However, further investigations with commensal-depleted or germ-free mice are necessary to validate this hypothesis.

Furthermore, one surprising finding is the remarkably high isotope labeling rate in the spleen. We observed that, of all the tissues examined, the spleen exhibited the highest labeling with NMN isotopes, regardless of whether NMN was administered through IP injection or gavage (Figure 4 and Figure 5). We also observed high nicotinamide and NR-derived NAD^+^ biosynthesis from the isotopically labeled NMN in the spleen. One possible explanation is that a population of cells, such as immune cells, migrates from the GI, where labeling is highest, to the lymphatic system and spleen. While still speculative, we are intrigued by the possibility that the isotope data could be explained by the nutrient-stimulated organ tropism of cells from the GI upon NMN administration.

While this study provides valuable insights into the metabolism and distribution of NMN-derived NAD^+^ in young, male C57BL/6J mice, there are several limitations that should be acknowledged. Firstly, this study did not examine the complete isotopic profiles and quantities in blood and tissues at all time points; therefore, certain crucial changes may have been overlooked. Secondly, the findings may not be applicable to female mice or mice of different ages or strains, and further research is necessary to determine the generality of these findings. Thirdly, the order of tissue collection and delay in harvesting may have impacted the absolute amounts and labeling of NAD^+^ measured, particularly in brown adipose tissues and brains, which were harvested last. Also, the limited time allowed for tissue collection prevented precise measurements of all muscles and individual tissues in the remaining carcass. Future experiments could benefit from longer treatment times to better understand the turnover of NAD^+^ in peripheral tissues, as well as targeted experiments to study NAD^+^ turnovers in peripheral tissues such as skin, connective tissues, and bones.

## 4. Materials and Methods

### 4.1. Isotopic NMN Synthesis

The synthesis of isotopic NMN involves four major steps: preparation of ^13^C labeled ribofuranoside tetra-acetate and ^18^O labeled nicotinamide; synthesis of ^13^C,^18^O-labelled nicotinamide riboside; and phosphorylation with ^18^O labeled H_2_O to produce NMN (See Appendix A).

1-^13^C ribofuranoside tetra-acetate was synthesized by following a reported method [33]. D-ribose was first converted to methyl ribofuranosides via methanolysis in anhydrous methanol containing concentrated sulfuric acid. The acetylation step can convert ribofuranosides to *β/α*-D-ribofuranose 2,3,5-triaacetate in the presence of acetic anhydride with pyridine as the basic condition. The acetolysis was carried out in a mixture of acetic anhydride and concentrated sulfuric acid to afford *β*/*α*-D-ribofuranose 1,2,3,5-tetraacetate, and *β*-isomer can be isolated via recrystallization from ethyl ether. ^1^H NMR (500 MHz, Chloroform-*d*) δ 6.35 (s, 1H), 5.99 (s, 1H), 5.36 (d, *J* = 6.2 Hz, 2H), 5.25 (ddt, *J* = 9.4, 6.7, 4.0 Hz, 1H), 4.45 (t, *J* = 3.4 Hz, 1H), 4.41–4.31 (m, 3H), 4.22 (dd, *J* = 12.2, 4.0 Hz, 1H), 4.16 (dd, *J* = 12.0, 5.4 Hz, 1H), 2.13 (d, *J* = 3.5 Hz, 7H), 2.10 (dd, *J* = 8.9, 3.9 Hz, 15H).

^18^O-Nicotinamide was prepared by following a reported procedure [34] with modifications. In brief, 3-cyanopyridine (0.936 g, 9.0 mmol) was added to 0.45 mL H_2_^18^O in a dry and sealed vial. The reaction mixture was heated at 70 °C for 4 h and stirred at 40 °C for 12 h. Water was then removed under reduced pressure, and the crude product was purified with flash chromatography on silica. Elution with hexane/acetone/methanol (3:1:0.02) gave pure nicotinamide-^18^O as a white solid (1.05 g, yield: 94.1%). ^1^H NMR (500 MHz, Methanol-*d*_4_) δ 9.04 (d, *J* = 2.4 Hz, 1H), 8.73–8.68 (m, 1H), 8.30 (dd, *J* = 7.8, 2.1 Hz, 1H), 7.56 (dd, *J* = 8.0, 5.0 Hz, 1H). ^13^C NMR (126 MHz, Methanol-*d*_4_) δ 168.39, 151.44, 148.07, 135.92, 130.04, 123.71.

^13^C,^18^O-labelled nicotinamide riboside was synthesized based on our previously reported method [35]. The synthesis of riboside was initiated by preparation of an intermediate ^13^C-β-D-ethyl nicotinate riboside 2,3,5-triaacetate via coupling synthesized ^18^O-ethyl nicotinate with ^13^C-β-D-ribofuranose 1,2,3,5-tetraacetate. ^1^H NMR (500 MHz, Deuterium Oxide) δ 9.63 (dt, *J* = 3.5, 1.7 Hz, 1H), 9.33–9.25 (m, 1H), 9.01 (dt, *J* = 8.1, 1.6 Hz, 1H), 8.31 (dd, *J* = 8.1, 6.3 Hz, 1H), 6.45 (d, *J* = 4.5 Hz, 0.5H), 6.10 (d, *J* = 4.5 Hz, 0.5H), 4.53 (ddd, *J* = 12.4, 5.7, 2.7 Hz, 2H), 4.39 (q, *J* = 4.4 Hz, 1H), 4.08 (dd, *J* = 12.9, 2.9 Hz, 1H), 3.93 (dd, *J* = 12.9, 3.6 Hz, 1H). ^13^C NMR (126 MHz, Deuterium Oxide) δ 165.75, 145.65, 142.63, 140.39, 128.40, 128.38, 123.40, 120.87, 118.35, 115.83, 107.89, 103.07, 102.99, 102.37, 102.35, 100.09, 99.93, 99.77, 98.31, 98.28, 96.61, 93.77, 87.69, 80.13, 77.60, 77.28, 69.80, 69.77, 60.19, 56.72. ^19^F NMR (471 MHz, Deuterium Oxide) δ −81.16.

To synthesize the [^18^O-phosphoryl-^18^O-carbonyl-^13^C-1-ribosyl] NMN, phosphoryl chloride (156 µL, 1.66 mmol) was added dropwise to the solution of ^13^C,^18^O-labelled nicotinamide ribose (226 mg, 0.56 mmol) in triethylphosphate (5 mL). The reaction mixture was stirred at 4 °C overnight and added to H_2_^18^O (100 µL). After stirring for another 30 min, water (5 mL) was added, and the pH value of the reaction mixture was adjusted to ~7.0 by adding triethylamine. The mixture was subsequently extracted with EtOAc (5 mL ×2), and the water layer was collected. Water was then removed under reduced pressure to give the crude product. The resulting mixture was subjected to reverse silica gel chromatography and eluted with water. The fractions containing ^18^O-NMN were combined and lyophilized to give an oily residue. The oily residue was then precipitated with isopropanol to provide pure tri-isotope-labeled NMN (127 mg, yield: 66.5%). ^1^H NMR (500 MHz, Deuterium Oxide) δ 9.48 (dt, *J* = 3.4, 1.6 Hz, 1H), 9.30 (ddd, *J* = 6.5, 3.3, 1.6 Hz, 1H), 9.00 (dt, *J* = 8.2, 1.6 Hz, 1H), 8.32 (dd, *J* = 8.1, 6.3 Hz, 1H), 4.66 (q, *J* = 2.5 Hz, 1H), 4.58 (td, *J* = 5.1, 3.5 Hz, 1H), 4.46 (td, *J* = 5.1, 2.6 Hz, 1H), 4.33 (ddd, *J* = 12.0, 4.4, 2.4 Hz, 1H), 4.17 (ddd, *J* = 12.0, 5.0, 2.2 Hz, 1H). ^31^P NMR (202 MHz, Deuterium Oxide) δ −0.23.

### 4.2. Animal Studies

For in vivo studies, male C57BL/6N mice purchased from Charles River Laboratories (Wilmington, MA, USA) were housed in polycarbonate cages under a 12-h-light/-dark cycle with free access to food and water. Mice were randomly assigned and received either an IP injection or oral gavage of 500 mg/kg of unlabeled NMN or [^18^O-phosphoryl-^18^O-carbonyl-^13^C-1-ribosyl] NMN dissolved in PBS, or only PBS in the control group. After 2, 4 or 6 h, these mice were subjected to cardiac puncture for blood collection and sacrificed. Blood was collected into BD Vacutainers containing EDTA and stored at −80 °C. Liver, kidney, pancreas, brain, muscle (quadriceps), spleen, lung, heart, stomach, epididymal white adipose, brown adipose tissue, peritoneal membrane, and peripheral tissue, which is the remaining carcass after removing all the above-mentioned internal organs, were harvested, weighed and immediately frozen in liquid nitrogen, then stored at −80 °C until analyses. Intestines were also harvested and cleaned, then divided into 4 sections of equal length before being snap-frozen. All procedures were approved by the Institutional Animal Care and Use Committee of Weill Cornell Medicine.

### 4.3. NAD^+^ and Nicotinamide Extraction and Quantification

For NAD^+^ analysis, ~100 mg of frozen tissue was pulverized in liquid nitrogen and homogenized in 7% perchloric acid by sonication (in the case of blood, perchloric acid was added directly to thawed aliquots), then the solution was neutralized and subjected to NAD^+^ measurement. The tissue NAD^+^ levels were measured using a cycling assay we developed and described [6]. In short, tissue extracts were combined with l-lactate, lactate dehydrogenase (LDH), diaphorase, and resazurin. NAD^+^ reacts with lactate catalyzed by LDH to form NADH, which then binds to the enzyme diaphorase to reduce the dye resazurin to form the reduced dye resorufin. Formation of resorufin is read by fluorescence (excitation at 530 nm, emission at 580 nm) with a plate reader. Standards of known NAD^+^ concentration were used to generate a standard curve. Final NAD^+^ concentrations in tissues were calculated and expressed as pmol/mg tissue. To calculate total NAD^+^ content, NAD^+^ concentrations (pmol/mg) in specific tissues were multiplied by the total tissue weights and reported as nmol. Mouse blood volumes are taken from those reported [36] to estimate total NAD^+^ in blood. Total amount of NAD^+^ in other tissues (remaining carcass after removal of Table 1 organ tissues) was determined by hind leg NAD^+^ content multiplied by carcass weight.

NAD^+^ concentration and total NAD^+^ content in major tissues of 10-week-old male C57BL/6N mice 4 h after administration with an IP injection or oral gavage of 500 mg/kg NMN dissolved in PBS, or PBS as vehicle control. NAD^+^ concentrations are expressed as mean (SD), pmol/mg tissue weight. Total NAD^+^ contents are expressed as mean (SD), nmol. *n* = 4 per group. * represents *p* < 0.05 when compared to the control. Fold represents the fold change of total NAD^+^ content over mean of controls. Total indicates the sum value of whole-body NAD^+^ content from all measured organs and remaining carcass.

To quantify blood nicotinamide levels, the extractions from blood were injected into a Hitachi Elite Lachrom HPLC system (Hitachi, Tokyo, Japan) equipped with a Diode Array Detector L-2450, using an EC 250/4.6 Nucleosil 100-5 C18 column. The C18 column was eluted with 20 mM ammonium acetate at 1 mL/min for 25 min, then with 20 mM ammonium acetate and 20% methanol for 20 min. Nicotinamide and NAD^+^ were characterized by their peaks at 260 nm. Nicotinamide concentrations were calculated based on the area-under-curves at 260 nm of known concentration standards and the samples. The nicotinamide and NAD^+^ peaks were also collected and lyophilized for further LC–MS analysis.

### 4.4. Mass Spectrum Analysis

To determine the mass of nicotinamide and NAD^+^, the lyophilized nicotinamide and NAD^+^ fractions were diluted and injected into Waters^®^ ACQUITY UPLC^®^ System (Milford, MA, USA) equipped with ACQUITY UPLC^®^ BEH C18 Column. Heights of each NAM and NAD^+^ *m*/*z* peak in positive ion mode were quantified using ImageJ and calculated in percentage relative to the height of a *m*/*z* = 122 or *m*/*z* = 664 peak. According to the prediction of NAD^+^ structure by ChemDraw^®^ 19.1.0.8, for an unlabeled NAD^+^ molecule with a molecular weight of 664, the *m*/*z* distribution is 664 (100.0%), 665 (23.6%), and 666 (6.1%), and we used this percentage to adjust for peak heights. For the 666 peak, the height was adjusted by subtracting 6.1% of the height of the 664 peak. For the 667 peak, the height was subtracted by 23.6% of the 666 peak. For the 669 peak, the height was subtracted by 6.1% of the height of the 667 peak.

### 4.5. Data Analysis

Data were expressed as mean ± S.D. One-way analysis of variance and Dunnett’s multiple comparisons tests were used to detect statistically significant differences between groups and were conducted using GraphPad Prism 9.5 (Boston, MA, USA). *p* values less than 0.05 were considered significant.

## 5. Conclusions

In conclusion, the findings from this study offer valuable and novel insights into the dynamics of whole-body NAD^+^ metabolism in mice. By employing triple-isotope-labeled NMN, we gained a clear understanding of the in vivo conversion process from NMN to NAD^+^. Our data revealed that the majority of absorbed NMN undergoes degradation to nicotinamide and NR before being utilized for NAD^+^ synthesis. Only a small fraction of the administered NMN is incorporated intact into NAD^+^, primarily in the kidney and white adipose tissues. Furthermore, the observed differences in NAD^+^ enhancement in the liver and the relatively low percentage of isotope labeling suggest a possible systematic redistribution of NAD^+^ throughout the body following NMN administration, particularly after oral intake. Overall, this study represents a crucial step for future in-depth investigations into NAD^+^ metabolism across various tissues, thereby paving the way for potential applications of NMN in human health.

## Figures and Tables

**Figure 1 ijms-24-11114-f001:**
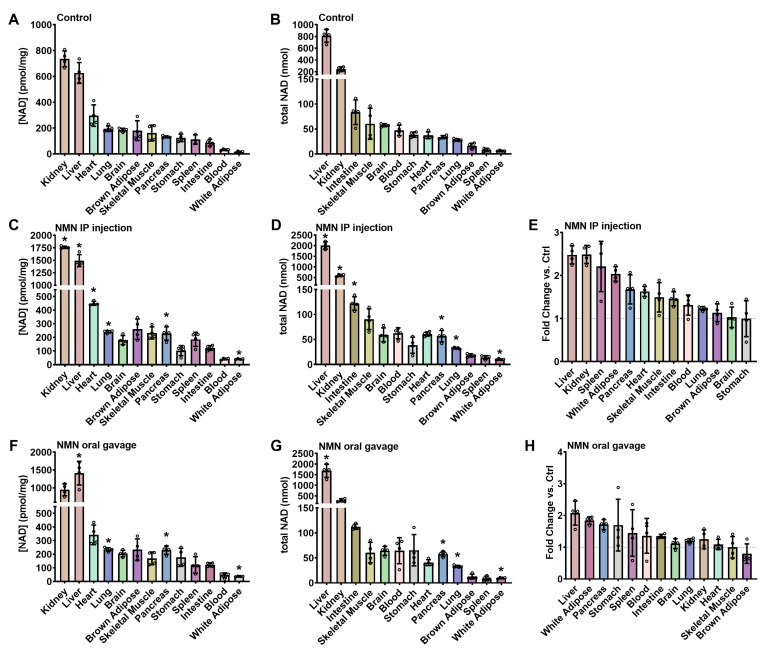
NAD^+^ distribution in mice tissues 4 h following NMN treatment. Male C57BL/6N mice at 10 weeks old received either an IP injection or oral gavage of 500 mg/kg NMN dissolved in PBS or only PBS in the control group, then were sacrificed at 4 h post-treatment. Tissue masses were recorded, and NAD^+^ concentrations were measured to calculate total NAD^+^ content. (**A**,**C**,**F**) show the NAD^+^ concentrations (pmol/mg) in Control, IP-injected, or oral-gavaged groups; tissues were ranked in order of their Control NAD^+^ concentrations; (**B**,**D**,**G**) show the total tissue NAD^+^ contents (nmol) in Control, IP-injected, or oral-gavaged groups; tissues were ranked in order of their total NAD^+^ amounts in Control; and (**E**,**H**) show treatment-over-control fold changes in NMN IP-injected and NMN oral-gavaged groups; tissues were ranked in order of their fold changes. Data are expressed as mean ± SD with individual data points, *n* = 4 per group. * indicates *p* < 0.05 when compared to the control.

**Figure 2 ijms-24-11114-f002:**
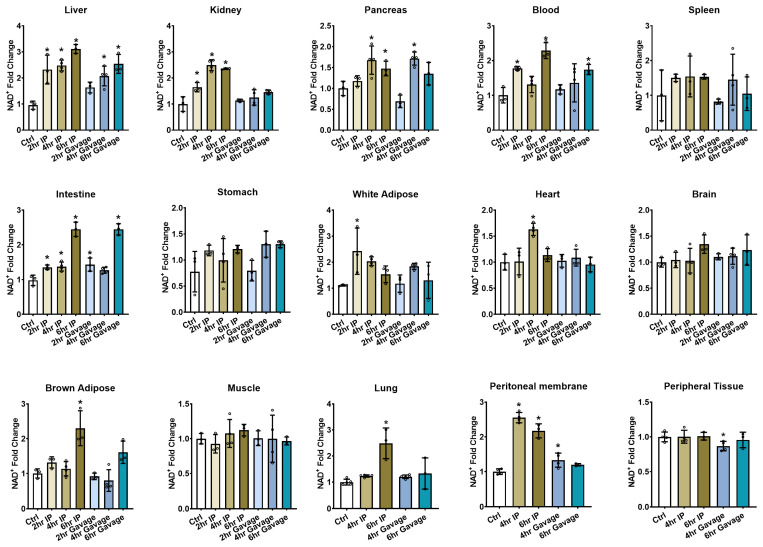
NMN effect on tissue NAD^+^ levels over time. Male C57BL/6N mice at 10 weeks old received either an IP injection or oral gavage of 500 mg/kg NMN dissolved in PBS, or only PBS in the control group, and were sacrificed at 2, 4, and 6 h post-treatment. Tissue NAD^+^ concentrations were measured and plotted as fold changes compared to the control mean. Data are expressed as mean ± SD with individual data points, *n* = 3–4 per group. * indicates *p* < 0.05 when compared to the control.

**Figure 3 ijms-24-11114-f003:**
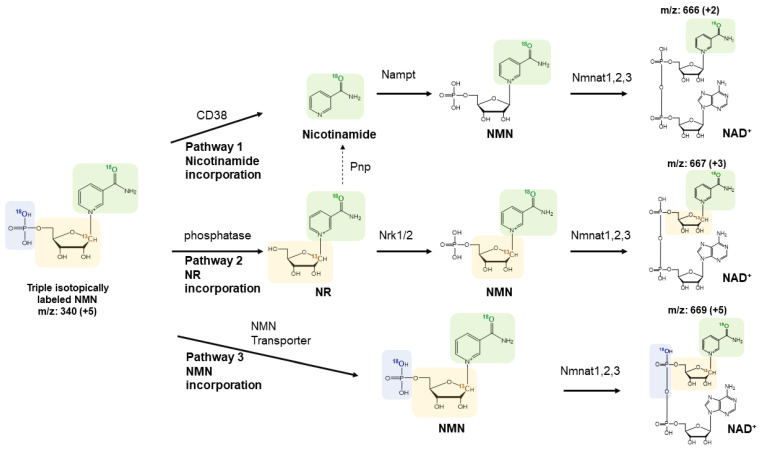
Triple isotope labeled NMN structure and potential pathways to NAD^+^. The scheme shows three potential pathways in which isotopically labeled NMN is metabolized into NAD^+^. In pathway 1, NMN is degraded into nicotinamide extracellularly or converted to NR, then degraded intracellularly into nicotinamide by Pnp, resulting in the production of ^18^O-labeled NAD^+^ (*m*/*z* = 666, +2). In pathway 2, NMN is dephosphorylated into NR before entering NAD^+^ biosynthesis, resulting in the production of ^18^O, ^13^C-labeled NAD^+^ (*m*/*z* = 667, +3). In pathway 3, the whole NMN molecule is incorporated into NAD^+^, producing newly synthesized NAD^+^ (*m*/*z* = 669, +5).

**Figure 4 ijms-24-11114-f004:**
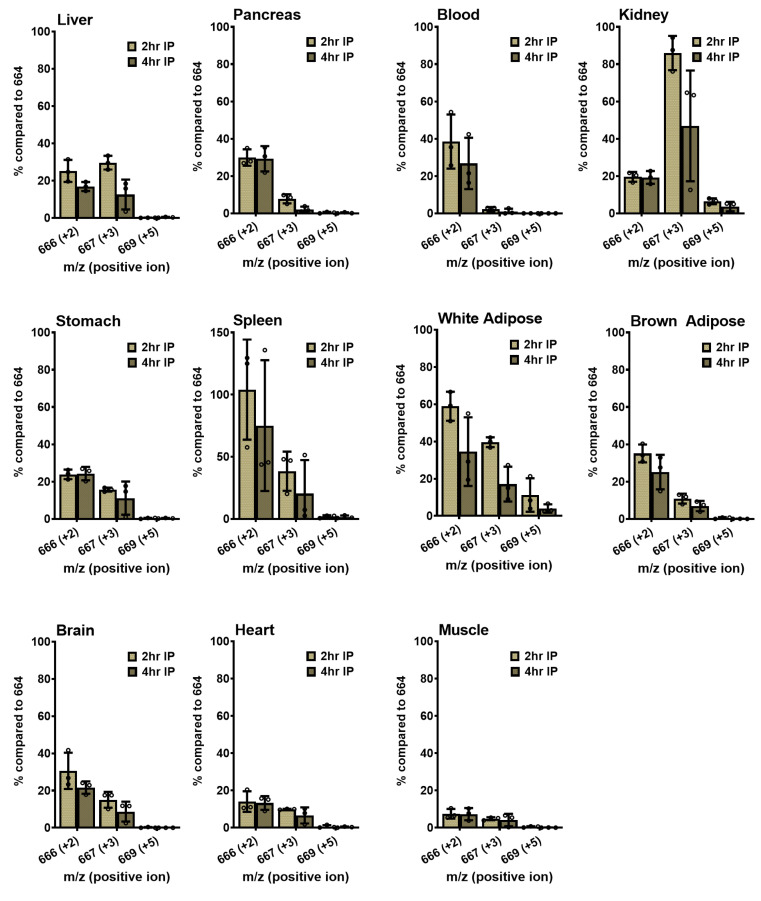
Triple isotope-labeled NMN induces NAD^+^ synthesis following IP injection in tissues. Of isotopically labeled NMN, 500 mg/kg were IP injected into 10-week-old male C57BL/6N mice, and the animals were sacrificed after 2 or 4 h. Their tissue NAD^+^ was extracted and isolated, then subjected to LC–MS analysis to examine their molecular mass. The levels of the 666, 667, and 669 peaks were quantified relative to the 664 peak and expressed as percentages compared to 664. Data are shown as mean ± SD with individual data points, *n* = 3 per group.

**Figure 5 ijms-24-11114-f005:**
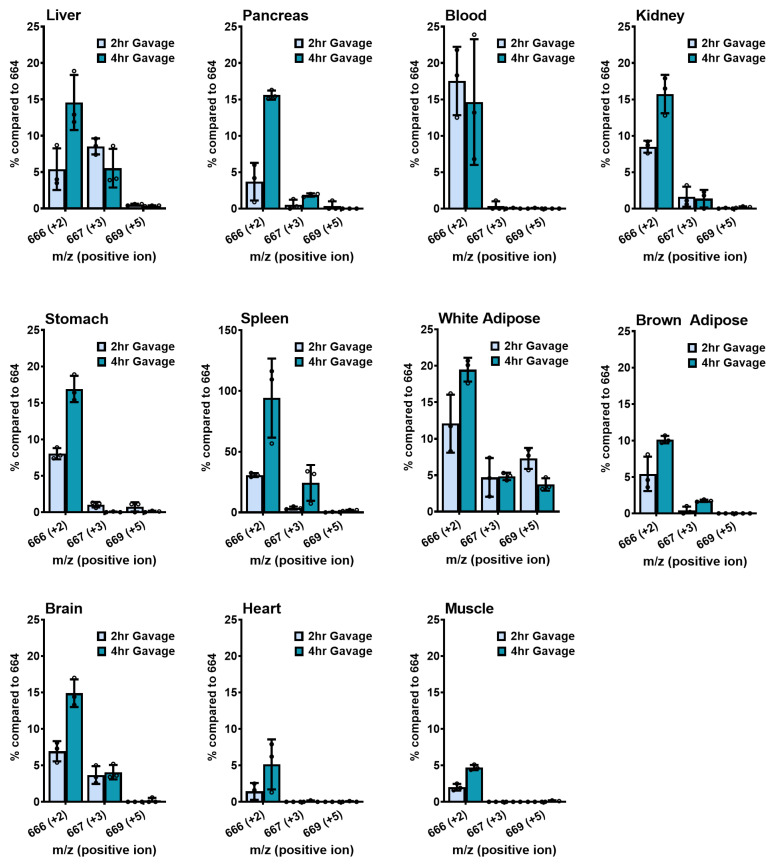
Triple isotope-labeled NMN induces NAD^+^ synthesis following oral gavage in tissues. Of isotopically labeled NMN, 500 mg/kg were orally gavaged into 10-week-old male C57BL/6N mice, and the animals were sacrificed after 2 or 4 h. Their tissue NAD^+^ was extracted and isolated, then subjected to LC–MS analysis to examine their molecular mass. The levels of *m*/*z* = 666, 667, and 669 intensities were quantified relative to the 664 peak and expressed as percentages compared to 664. Data are shown as mean ± SD with individual data points, *n* = 3 per group.

**Figure 6 ijms-24-11114-f006:**
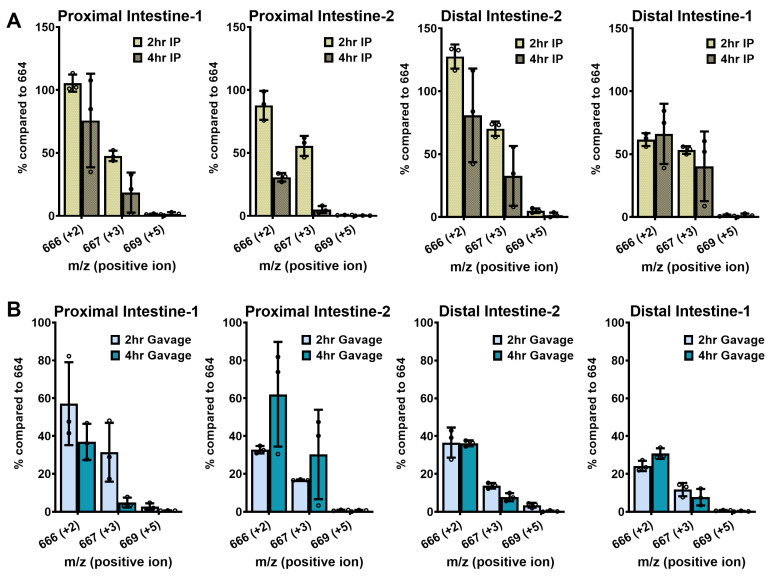
Triple-labeled NMN induces NAD^+^ synthesis in intestinal fractions. Of isotopically labeled NMN, 500 mg/kg were either (**A**) IP injected or (**B**) orally gavaged into 10-week-old male C57BL/6N mice, and the animals were sacrificed after 2 or 4 h. Their intestines were evenly cut into four sections, and their NAD^+^ was extracted and isolated, then subjected to LC–MS analysis to examine their molecular mass. The levels of *m*/*z* = 666, 667, and 669 peaks were quantified relative to the 664 peak and expressed as a percentage compared to 664. Data are shown as mean ± SD with individual data points, *n* = 3 per group.

**Figure 7 ijms-24-11114-f007:**
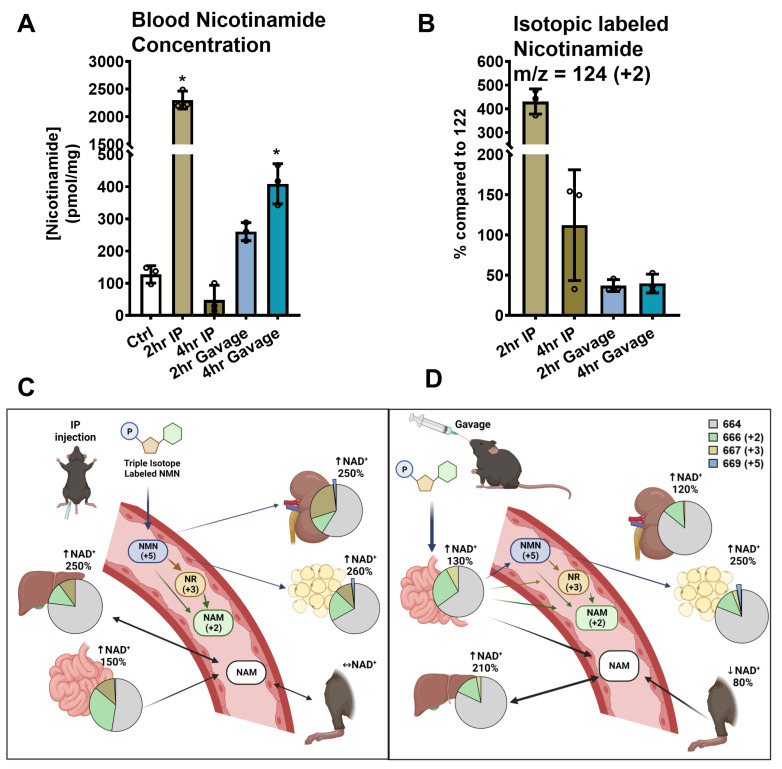
Isotopic labeling in blood nicotinamide. Following IP injection or oral gavage of 500 mg/kg triple isotope labeled NMN for 2 or 4 h, the blood nicotinamide was extracted and quantified by its area-under-curve at the 260 nm peak on HPLC chromatograms. (**A**) shows the concentrations of nicotinamide calculated based on the area under the curve for known nicotinamide standards and normalized by the original blood weight (mg) injected into the HPLC column. (**B**) shows the percentage of isotope-labeled nicotinamide peaks, *m*/*z* = 124 (+2), in relation to the *m*/*z* = 122 peak that was set to be 100%. Data are shown as mean ± SD with individual data points, *n* = 3. (**C**,**D**) Schematic illustrating the NMN metabolism following (**C**) IP injection and (**D**) oral gavage. The pie charts represent the proportion of NAD^+^ with no isotope labels (gray), with labeled nicotinamide (+2, green), with labeled NR (+3, yellow), and with labeled NMN (+5, blue) in each tissue at 4 h. Black arrow indicates the movement of unlabeled nicotinamide, and blue arrow represents the movement of labeled NMN. * indicates *p* < 0.05.

**Table 1 ijms-24-11114-t001:** Tissue NAD^+^ values in Ctrl or NMN-treated C57/B6 mice 4 h after administration with an IP injection or oral gavage of 500 mg/kg NMN dissolved in PBS, or PBS as vehicle control. NAD^+^ [concentrations] are expressed as mean (SD) pmol/mg tissue weight. Total NAD^+^ contents (c) are expressed as mean (SD), nmol. *n* = 4 per group. * represents *p* < 0.05 when compared to control. Fold represents the fold change of total NAD^+^ content over the mean of controls. Total indicates the sum value of whole-body NAD^+^ content from all measured organs and the remaining carcass.

Tissue	[NAD^+^]	[NAD^+^]ip	[NAD^+^]g	NAD^+^c	NAD^+^cip	NAD^+^cg	Foldip	Foldg
**Kid**	736(61)	1761(23) *	955(166)	242(40)	601(51) *	302(71)	2.5(0.2)	1.2(0.3)
**Liver**	621(81)	1493(121) *	1413(333) *	812(109)	2012(172) *	1687(307) *	2.5(0.2)	2.1(0.4)
**Heart**	297(83)	450(15) *	342(73)	37(7)	61(4) *	41(6)	1.6(0.1)	1.1(0.2)
**Lung**	195(21)	241(9) *	235(13) *	28(2)	32(2) *	33(2) *	1.2(0.1)	1.2(0.1)
**BAdip**	181(76)	260(75)	233(79)	16(6)	18(3)	13(5)	1.1(0.2)	0.8(0.3)
**Skele Musc**	162(61)	232(45)	169(47)	61(31)	90(21)	61(21)	1.5(0.3)	1.0(0.3)
**Brain**	185(14)	180(34)	205(24)	58(3)	59(14)	64(9)	1.0(0.2)	1.1(0.2)
**Pancr**	132(8)	229(49) *	230(31) *	34(3)	57(12) *	58(5) *	1.7(0.3)	1.7(0.2)
**Stoma**	124(29)	103(37)	177(66)	39(5)	38(16)	66(31)	1.0(0.4)	1.7(0.8)
**Spleen**	112(36)	185(52)	121(60)	6(5)	14(4)	9(5)	2.2(0.6)	1.4(0.7)
**Intest**	87(25)	122(18)	123(14)	84(25)	122(13) *	113(5)	1.5(0.2)	1.3(0.1)
**Blood**	32(5)	41(6)	45(18)	47(10)	62(11)	64(26)	1.3(0.2)	1.4(0.6)
**WAdip**	15(6)	40(3) *	37(2) *	4(1)	10(1) *	10(1) *	2.6(0.3)	2.5(0.3)
**Carcs**	171(14)	161(15)	139(11) *	3178 (325)	3048 (404)	2644 (200) *	1.0(0.1)	0.8(0.1)
**Total**				4638(582)	6225(726)	5163(693)		

## Data Availability

All relevant data are available from the corresponding author upon reasonable request.

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
