# Peer review of "Triple-Isotope Tracing for Pathway Discernment of NMN-Induced NAD^+^ Biosynthesis in Whole Mice"

_ijms, 2023, doi:10.3390/ijms241311114_

Round 1
Reviewer 1 Report
The present manuscript re-examines the fate of oral and IP NMN supplements in mice. The quality of the data and choice of isotope tracer are good. Largely the research findings align with prior literature. Concerns:
1. Claims in the abstract that some well-studied areas remain unclear despite prior work that is quite close to the present work are unfounded. The authors would be better served simply highlighting the importance of rigorous evaluation of this important topic from multiple complementary angles to ascertain robust answers of high confidence for the field. The paper itself largely handles this appropriately, the abstract is the most problematic part.
2. Claims of systemic pool redistribution (including in the title) are inadequately supported. Indeed, I really cannot figure out what data drove the authors to this somewhat radical conclusion. The full isotopic profiles and quantities in blood and tissue NAM are not readily accessible. The authors did not seem to adequately consider the possibility that unlabeled NAD can accumulate in response to labeled NMN supplementation in part because the resulting labeled NAD partially fulfills catabolic requirements (e.g. PARP, sirtuins), leading to unlabeled buildup. The authors also did not adequately consider the nicotinic acid pathway, which has emerged as crucial for NR processing and is likely also contributing here, with their tracer choice lacking for this particular route. A more conservative title and lack of under-substantiated claims in the abstract are essential; a speculative paragraph might be appropriate in the Discussion, hopefully with more clear arguments on this front than the current manuscript.
Author Response
We thank the reviewer for the comments!
Regarding Concern 1:
To avoid overstating the knowledge gap in prior works, we revised the abstract and modified the language to “However, despite these extensive efforts, our understanding and characterization of their whole-body pharmacodynamics, impact on NAD+ tissue distribution, and mechanism of action in various tissues remain incomplete.” To highlight the merit of rigorous evaluation, we made the following changes in the abstract: “In this study, we administered NMN via intraperitoneal injection or oral gavage, and conducted a rigorous evaluation of NMN's pharmacodynamic effects on whole-body NAD+ homeostasis in mice. To provide more confident insights into NMN metabolism and NAD+ biosynthesis across different tissues and organs, we employed a novel approach using triple-isotopically labeled [18O-phosphoryl-18O-carbonyl-13C-1-ribosyl]NMN. …Our results provide more comprehensive characterization of the NMN impact on NAD+ concentrations and absolute amounts in various tissues and the whole body.”
Regarding Concern 2:
We agree that the full isotopic profiles at all time points have not been completely measured in this study, and the evidence supporting the concept of "pool distribution" was not sufficiently robust. Therefore, we changed our title to: “Triple-Isotope Tracing for Pathway Discernment of NMN-Induced NAD+ Biosynthesis in Whole Mice” to be more conservative. We have also removed related claims “We also observed the redistribution of endogenous, unlabeled NAD+ and nicotinamide from intestines and other tissues upon gavage administration of NMN, resulting in a substantial increase of unlabeled NAD+ in metabolically active tissues. “from the Abstract; and deleted the claim “In addition, we show evidence of mobile pools of unlabeled nicotinamide that significantly contribute to the effects of NMN on NAD+ boosting and NAD+ isotope labeling in multiple tissues.” from Introduction.
We also made related changes throughout the Result and Discussion sections to clarify our argument on the redistribution theory. These are highlighted in blue in the manuscript. We included the following statement in the last paragraph of the discussion to highlight the limitations of this study: “Firstly, this study did not examine the complete isotopic profiles and quantities in blood and tissues at all time points, therefore, certain crucial changes that may have been overlooked.”
Regarding the nicotinic acid (NA) pathway, we concur with the reviewer that it plays a significant role in in vivo NAD+ metabolism, and incorporating a model such as commensal-depleted or germ-free mice to assess its contribution would be a crucial future step. However, based on previous publications, we do not believe that NA-induced NAD+ synthesis has a significant impact on the interpretation of our results. NAD+ synthesis through NA could potentially result in a minimum of 50% loss of the 18O-carbonyl label in our isotopically labeled NMN. In Reference 24 by Liu et al., they employed labeled NMN with 13C incorporated within the nicotinamide moiety, which remains with NA upon conversion. However, they observed less than 15% labeling in the liver after administering the labeled NMN through gavage, which aligns with our findings (12% labeling at 2 hours and 17% at 4 hours). This led us to believe that NA-induced NAD+ synthesis may not be a major contributor when NMN is administered orally via gavage. We have modified the related discussion paragraph for clarification.
About NAD+ catabolism, we believe that the tissues preferentially using labeled NAD+ instead of unlabeled NAD+ is not a likely scenario. Our study is based on the presumption that enzymes will utilize both labeled and unlabeled NAD+ in a similar manner, and that the rate of reduction in isotopically labeled NAD+ should reflect the rate of catabolism in whole NAD+ pool. (Similar presumptions have been used in other previous works including Reference 24, in which the authors used isotopically labeled NAD+ to determine its turnover rate). Therefore the buildup of the unlabeled NAD+ pool should primarily reflects the biosynthesis of NAD+ from unlabeled substrates.
Reviewer 2 Report
The manuscript is well written and merits publication.
I have a suggestion that they can move synthesis as supplementary information or if they would like to keep it as it is, a schematic illustration of the synthesis will help as visual presentation to the readers.
Author Response
We thank the reviewer for the comment.
To clarify the chemical synthesis, we now add a description in the method “The synthesis of isotopic NMN involves four major steps: preparation of 13C labeled ribofuranoside tetra-acetate and 18O labeled nicotinamide, synthesis of 13C,18O-labelled nicotinamide riboside and phosphorylation with 18O labeled H2O to produce NMN (See Supplement Scheme).” And provide a scheme for the isotopically labeled NMN synthesis steps in Supplementation.
Round 2
Reviewer 1 Report
acceptable revision